# StyleDreamer: Make Your 3D Style Avatar from a Single View with Consistency Score Distillation

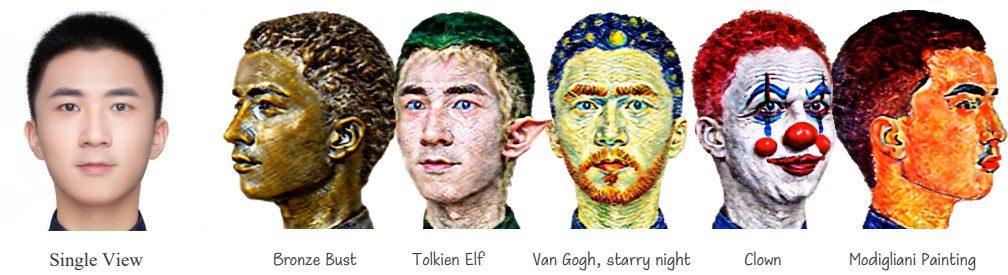

Figure 1: **One-to-Style:** We propose a new task, aiming at generating stylized 3D head avatars based on only one view. In this way, people can synthesize their avatars in different styles handily.

## Abstract

Recent generative methods have shown remarkable capabilities in producing stylized 3D head avatars. Nonetheless, current methods necessitate multi-view images for constructing 3D models, which limits their usage in the real world where most images are captured from a single angle. In this paper, we investigate a practical *One-to-Style* task, generating the 3D style avatar with a single view. The task presents two challenges: 1) **Content** consistency and 2) **Style** consistency across multiple views of the generated images. We propose a novel Consistency Score Distillation (CSD) to ensure consistent stylization across multiple views while preserving the identity of each view using the 3D GAN. In this method, the style distribution of the rendered images from all views is supervised to match the style of the given single view, based on the provided edit instruction. We have formulated a dataset for One-to-Style with in-the-wild face images and the most commonly used style. Experimental results show that our approach outperforms existing methods in terms of stability and quality, indicating its potential applications in the real world. Result videos can be found on the project website: https://one-to-style.github.io/.

## 1 Introduction

Recently, generative models (*e.g.*, diffusion (Ho et al., 2020; Song et al., 2020)) have made unprecedented advancements in high-quality image synthesis. With image-text alignment techniques (*e.g.*, CLIP (Radford et al., 2021)), generative models are able to produce high-fidelity, diverse and controllable images (Ho et al., 2022; Rombach et al., 2022). Meanwhile, NeRF (Mildenhall et al., 2020), a representation for 3D reconstruction in neural fields, indicates it is possible to generate 3D scenes with 2D images. With the continuous maturity of these generation technologies, the generation of 3D from text has become feasible. With these generation technologies, text-guided 3D generation has become achievable.(*e.g.*, DreamFusion (Poole et al., 2022)).

Creating 3D head avatars is an essential challenge with diverse applications, including animation (Ma et al., 2023), gaming (Zhao et al., 2023), virtual reality and augmented reality (Zhang et al., 2023c). Given a portrait image, recent 3D face reconstruction methods (Chan et al., 2022;

An et al., 2023; Zhang et al., 2023b; Gal et al., 2022b) are able to generate a 3D avatar. Despite promising results, these methods are unable to generalize to out-of-domain styles since they solely concentrate on one style during training. Consequently, producing personal avatars with diverse styles becomes unattainable using these methods.

In this paper, we introduce a practical *One-to-Style* task, which aims at generating 3D head avatars with a variety of styles based on only one view, as shown in Fig. 1. Specifically, given a portrait image that contains the front view of a head, and a textural prompt that specifies the style of interest, the task is to reconstruct the corresponding 3D head avatar. In this case, people can synthesize their personal avatars in different styles conveniently.

There are two distinct challenges in the proposed One-to-Style task: (1) **Content consistency**: Constructing a 3D head avatar using just one image makes it difficult to ensure the accurate creation of other perspectives. This can result in a generated 3D avatar that does not align with the intended representation of the image. To address this issue, it is necessary to generate unseen views that maintain consistency with the person in the single view. (2) **Style consistency**: Directly applying image editing to individual images can result in inconsistent stylization across multiple perspectives. Therefore, achieving consistent stylization across multiple views poses a significant challenge.

To maintain both style and identify of images from multiple views, we introduce a pipeline for generating stylized 3D head avatars based on a single view, dubbed as *StyleDreamer*. It consists of two components: (1) **3D-aware priors sampling**: we propose to use a 3D GAN to synthesize novel views that are consistent with the portrait image. Next, we adopt an empirical sampling strategy to select the reference images that benefit the following stylization. (2) **Consistency score distillation**: Considering the inconsistency of stylization made by per-frame editing, we equip the image editing method with score distilling-based loss. It progressively updates the 3D scene and aligns the editing through a 3D-aware representation (*e.g.*, NeRF (Mildenhall et al., 2020)). As shown in Fig. 1, our method produces high-fidelity stylized 3D head avatars from a single view given different editing instructions.

In summary, our main contributions are as follows:

- A new problem: *One-to-Style*. We build a benchmark dataset composed of portrait images in the wild, as well as style prompts including artistic, material, and character styles.
- A stylized 3D head avatar generation pipeline: *StyleDreamer*. It encourages the generation of consistency in both content and style via 3D-aware priors sampling and consistency score distillation, respectively.
- Experiments show that our method creates head avatars with rich detail in diversity styles.

## 2 RELATED WORK

**Text-to-2D generation.** Recently, with the development of vision-language models (Radford et al., 2021) and diffusion models (Sohl-Dickstein et al., 2015; Ho et al., 2020), great advancements have been made in text-to-image generation (T2I) (Zhang et al., 2023a). In particular, GLIDE (Nichol et al., 2021) introduces classifier-free guidance in T2I, facilitating the utilization of free-form prompts. Additionally, Imagen (Ho et al., 2022) adopts a pretrained and frozen large language model (Devlin et al., 2018; Brown et al., 2020) as the text encoder, further improving the image fidelity and image-text alignment. StableDiffusion (Rombach et al., 2022) is a particularly notable framework that trains the diffusion models on latent space, leading to reduced complexity and detail preservation. Meanwhile, some works are dedicated to spatial control (Voynov et al., 2023; Zhang et al., 2023d), concept control (Gal et al., 2022a; Ruiz et al., 2022), and adopting knowledge-based retrieval for out-of-distribution generation (Blattmann et al., 2022; Chen et al., 2023), etc. with the emergence of text-to-2D models, more fine-grained applications have been developed, including video generation (Singer et al., 2022; Ho et al., 2022), story visualization (Pan et al., 2022; Rahman et al., 2023), and text-guided image editing (Kim et al., 2022a; Brooks et al., 2023).

**Text-to-3D generation.** The success of the 2D generation is incredible. However, directly transferring the image diffusion models to 3D is challenging, due to the difficulty of 3D data collection. Recently, Neural Radiance Fields (NeRF) (Mildenhall et al., 2020; Barron et al., 2022) opened a new insight for the 3D-aware generation, where only 2D multi-view images are needed in 3D

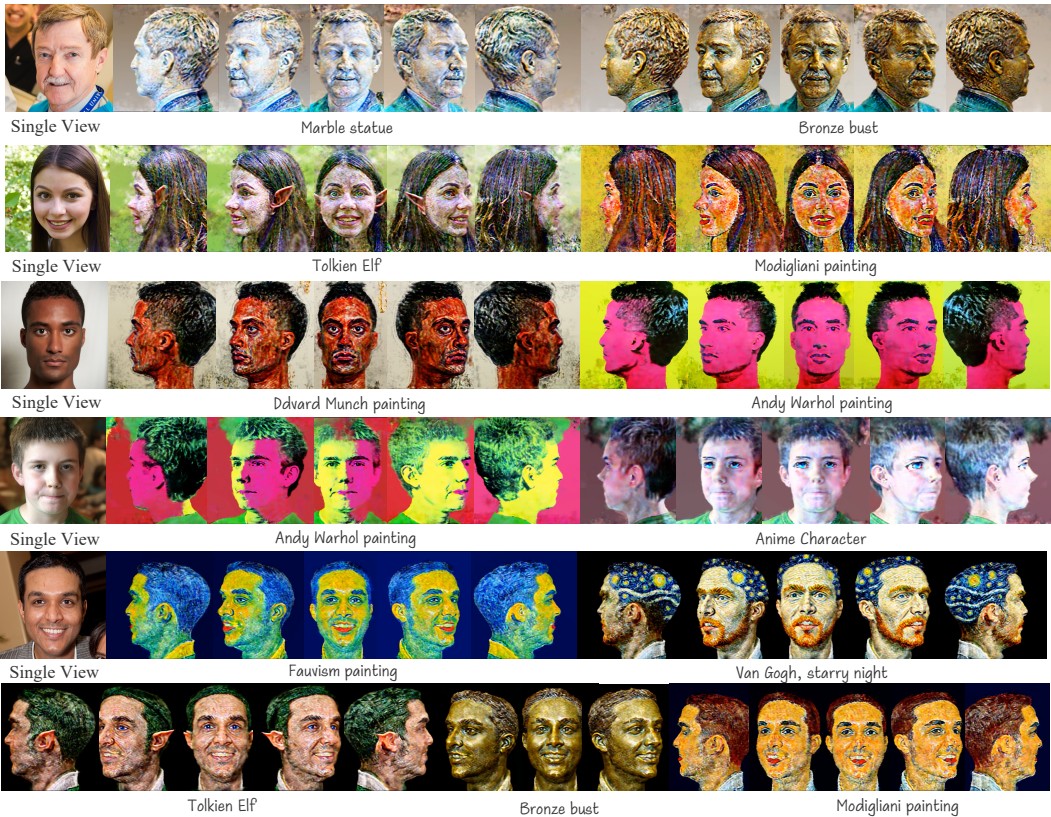

Figure 2: **Qualitative Results:** Given a front view portrait image, our method is able to generate 3D head avatars in diverse styles, including *artistic style*, like Van Gogh's Starry Night, *material style*, such as bronze bust, and even *character style* (Tolkien Elf). The evaluated images are randomly sampled from the FFHQ dataset (Karras et al., 2019).

scene reconstruction. Combining prior knowledge from text-to-2D models, several methods, such as DreamField (Jain et al., 2022), DreamFusion (Poole et al., 2022), and SJC (Wang et al., 2022), have been proposed to generate 3D objects guided by text prompt (Li et al., 2023). Additionally, numerous researches endeavor to improve this paradigm, such as effective guidance (Lin et al., 2023b), personalized generation (Raj et al., 2023), rendering speed (Nichol et al., 2022; Li & Kitani, 2023), 3D consistency (Hong et al., 2023; Seo et al., 2023), and high-fidelity (Metzer et al., 2022; Lin et al., 2023a; Wang et al., 2023b). Moreover, the recent advancement of text-to-3D models also inspired multiple applications, including text-guided scenes generation (Cohen-Bar et al., 2023; Höllein et al., 2023), text-guided avatar generation (Cao et al., 2023; Jiang et al., 2023), and text-guided 3d model editing (Haque et al., 2023; Kamata et al., 2023).

**3D head avatar generation and stylization.** At early stages, 3D head generation is based on statistic models, such as 3D Morphable Model (3DMM) (Blanz & Vetter, 1999) and FLAME model (Li et al., 2017). 3D-aware Generative Adversarial Networks (GANs) achieved incredible performance in 3D head synthesis, such as GRAF (Schwarz et al., 2020), pi-GAN (Chan et al., 2021), EG3D (Chan et al., 2022), and PanoHead (An et al., 2023), etc. Among them, EG3D (Chan et al., 2022) first introduces the tri-plane, a highly efficient 3D representation, for more view-consistent head synthesis. Improving tri-plane, PanoHead (An et al., 2023) proposes tri-grid representation to realize full heads synthesis in 360°. Meanwhile, many researches (Gal et al., 2022b; Abdal et al., 2023; Zhang et al., 2023b) explore domain adaptation of trained 3D GANs for generating 3D models with different styles. However, these methods require a large amount of data for supervised training and struggle to generalize well to out-of-domain avatars. Another line of work generates 3D head avatars by utilizing knowledge priors in 2D diffusion models, including DreamFace (Zhang et al., 2023c), HeadSculpt (Han et al., 2023). Despite producing promising results, these methods are not

able to generate unknown avatars. In contrast, our approach relies solely on a single-view image, generalizes well on various style avatars, and supports the user image-guided generation.

## 3 ONE-TO-STYLE PROBLEM

The application of customized stylized 3D head avatars is hindered by the demand for multi-view head images. We introduce a new task called One-to-Style where a stylized 3D head avatar is generated using only a front-view portrait and a style prompt. As a conditional generation task, the identity and style of the 3D head avatar are determined by the portrait and prompt, respectively. To evaluate the rendered head avatars from the single view, we introduce the dataset and evaluation metrics in this section.

**Benchmark dataset.** This dataset is used to evaluate the stylized 3D head avatar generation performance, which is composed of portrait images $\{\tilde{x}^i\}_{i=1}^{n_{\text{img}}}$ and style prompts $\{y^i\}_{i=1}^{n_{\text{sty}}}$. First, we collect $n_{img}$ portrait images from FFHQ (Karras et al., 2019). With the development of face-related tasks (Kumar et al., 2020; Zhang et al., 2022), numerous large-scale face image datasets (Wu et al., 2018; Karras et al., 2019; Karkkainen & Joo, 2021) have been proposed. As mentioned by Karkkainen & Joo (2021), the public face image datasets are strongly biased toward Caucasian faces, which limits the evaluation effect. So, we follow the race, gender, and age balance strategy in FairFace and build a tiny dataset from FFHQ, as basic. Meanwhile, considering the time-consuming of optimized-based methods, we randomly collect $n_{\text{img}} = 10$ front view portrait images with $1024^2$ resolution from the tiny dataset as an evaluation dataset. Second, we collect $n_{\text{sty}} = 15$ style prompts covering the most commonly used style from the state-of-the-art stylized generation methods (Gal et al., 2022b; Wang et al., 2023a; Haque et al., 2023; Zhang et al., 2023b), including artistic style (*e.g.*, Modigliani painting), material style (*e.g.*, marble statue) and character style (*e.g.*Tolkien Elf). Based on the proposed benchmark dataset, the generalization ability of methods to user input could be evaluated well. The more detailed analysis of benchmark dataset can be found in our appendix.

**Evaluation metrics.** Qualitative evaluation is necessary to evaluate the stylized generation. In this paper, we introduce two metrics, a *ID Score*, and a *Style Score*, to quantitatively evaluate the identity reservation and style transformation, respectively. The ID score measures the identity reservation using the identity similarity score by calculating the Adaface (Kim et al., 2022b) cosine similarity score between the given image and the corresponding rendered image. The style score measures the cosine similarity between the render images and the target style prompt, using a pre-trained CLIP (Radford et al., 2021). In this way, the proposed two metrics could evaluate the content and style consistency of generated avatars, respectively.

## 4 METHOD

In One-to-Style, a high-quality 3D avatar head is defined to be consistent with the given portrait image and style prompt simultaneously. Achieving this consistency involves overcoming two challenges: maintaining content consistency and ensuring style consistency. The former enforces the created 3D model that looks like the person (in the portrait image) when rendered from random angles. The latter requires a consistent stylization across multiple views. Therefore, generating high-quality 3D avatars depends on solving these two challenges at once.

To reconstruct high-quality stylized 3D head avatars, we design a novel generation pipeline, Style-Dreamer. As shown in Fig. 3, StyleDreamer encourages the generation of consistency in both content and style. Specifically, with the help of the pre-trained 3D GAN, we first reconstruct the person's 3D face and then synthesize consistent novel views, termed as **3D-aware priors sampling** (Sec. 4.2). Then, considering the inconsistency stylization by per-frame editing, we equip the image editing method with score distilling-based loss, dubbed **consistency score distillation** (Sec. 4.3). It progressively updates the 3D scene and aligns the editing through a 3D-aware representation (*e.g.*, NeRF). Next, we first introduce the preliminaries that form the basis of our method.

### 4.1 PRELIMINARY

**NeRF-based 3D representations.** Neural radiance field (NeRF) (Mildenhall et al., 2020; Müller et al., 2022) is a popular and important 3D representation. In particular, NeRF assumes that each

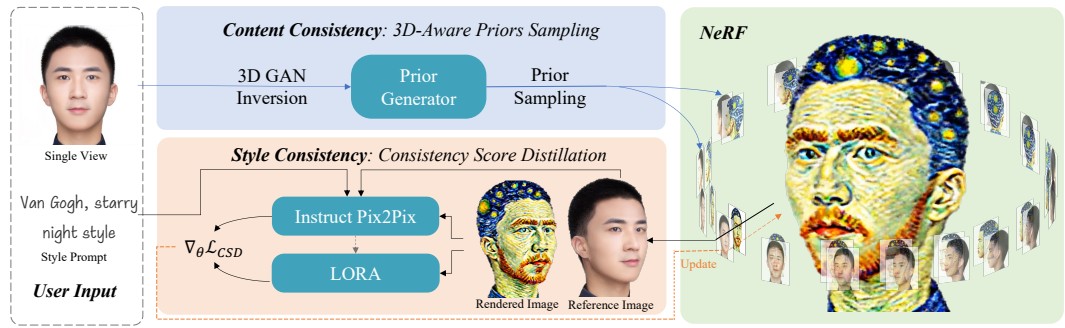

Figure 3: **StyleDreamer:** Our method generates a 3D head avatar whose content and style are consistent with the given portrait image and style prompts, respectively. (0) Given a front-view portrait image, novel views are synthesized by a 3D GAN, as reference images. (1) An image is rendered from NeRF at a training pose. (2) The rendered image is sent to InsturctPix2Pix, conditioned by corresponding reference image and style prompt, as well as the score of the rendered image distribution (estimated by LoRA) to compute the gradient of CSD. (3) The NeRF is updated by the gradient of CSD, and LoRA is also updated on the rendered image.

pixel in an image is rendered by a ray passing through the 3D scene. The per-pixel ray $r$ is calculated by the camera pose $c$. The color of this ray is determined by the weighted sum of the sampling points' color along the ray. Furthermore, the color and density of a 3D point are modeled by a multi-layer perception, parameterized by $\theta$, which takes position and viewing direction as input. The typical training process of NeRF involves selecting a subset of rays $r$, rendering the color $\hat{C}$ along this ray, and computing a loss relative to captured pixel color $\mathcal{L}_\theta(\hat{C}, C)$. Given enough multi-view images and corresponding camera poses, NeRF is capable of representing extremely complex scenes.

**Diffusion-based generation.** Generative modeling with diffusion models (Rombach et al., 2022; Ho et al., 2022) consists of a forward process $q$ that gradually adds Gaussian noise to the input $x_0 \sim p_{data}(x)$, and a reverse process $p$ which gradually denoises from the Gaussian noise $x_T \sim \mathcal{N}(0, \mathbf{I})$. The training process of diffusion models is to predict the added noise:

$$\mathcal{L}_{\text{Diff}}(\phi) = \mathbb{E}_{x_0, t, \epsilon}[w(t)||\epsilon_\phi(x_t; t) - \epsilon||_2^2], \ x_t = \alpha_t x_0 + \sigma_t \epsilon, \tag{1}$$

where $w_t$ is a weighting function that depends on the timestep t, and $\alpha_t$ is a pre-defined constant.

Using pre-trained text-to-2D diffusion models, Wang et al. (2023b) proposed variational score distillation (VSD) for high-quality text-to-3D generation. VSD treats the 3D scene as a random variable instead of a single point as in SDS (Poole et al., 2022) and optimizes a distribution of 3D scenes that aligns with the one defined by the pre-trained diffusion model. VSD realizes this optimization via simulating an ODE (Lu et al., 2022), which involves the score function of noisy real images and noisy rendered images. Additionally, VSD uses a LoRA (Low-rank adaptation (Hu et al., 2021)) of the pre-trained model $\epsilon_\phi(x_t; y, t, c)$ to learn the distribution of noisy rendered images. Formally, the gradient of VSD loss is given by

$$\nabla_\theta \mathcal{L}_{\text{VSD}}(\theta) = \mathbb{E}_{t, \epsilon, c}[w(t)(\epsilon_{\text{pretrain}}(x_t; y, t) - \epsilon_\phi(x_t; y, t, c))\frac{\partial x}{\partial \theta}]. \tag{2}$$

## 4.2 CONTENT CONSISTENCY: 3D-AWARE PRIORS SAMPLING

To retain the content consistency, the generated 3D model should have the same identity as the person depicted in the portrait image. The main focus of this problem is to synthesize novel views that match the portrait image. We choose the pre-trained 3D GAN model to achieve this that has been trained on a collection of human faces (Karras et al., 2019). Then, we employ GAN inversion to reconstruct the 3D face, allowing us to generate novel views that preserve content consistency. These generated novel views can serve as reference images in further text-image conditioned 3D generation.

**Single-view 3D GAN Inversion.** We first use PanoHead (An et al., 2023) for full-head reconstruction from a single-view portrait $\tilde{x}$. Specifically, we perform optimization to find the suitable latent

code $z_{\tilde{x}}$ for the target image using pixel-wise $\mathcal{L}_2$ loss and image-level LPIPS loss (Zhang et al., 2018a). Then, we perform pivotal tuning inversion (PTI) (Roich et al., 2022) to fine-tune the generator parameters with a fixed optimized latent code $z_{\tilde{x}}$, to further improve the reconstruction. In this way, we obtain a prior generator with fixed latent code $z_{\tilde{x}}$, termed as $g_{\text{prior}}(\cdot)$. Benefiting from the the development of full head reconstruction, we could get a high-fidelity novel-view synthesis in $360°$, including a large pose and back head. Given a camera parameter $c^i$, the novel view $\hat{x}^i$ can be rendered as below:

$$\hat{x}^i = g_{\text{prior}}(c^i; z_{\tilde{x}}). \tag{3}$$

**Experience-guided priors sampling strategy.** Random camera and light sampling are commonly used in 3D generation (Poole et al., 2022; Lin et al., 2023a; Wang et al., 2023b). However, performing online sampling of camera pose and rendering novel views at each iteration is time-consuming. To address this, we opt to synthesize the novel view offline instead. In practice, sampling novel view with different camera pose more will contribute to a better reconstruction performance. However, in stylized 3D head generation, it also introduces more inconsistency, leading to geometry and texture distortion. To balance the trade-off between priors information and stylized consistency, we introduce an experience-guided priors sampling strategy. Specifically, we follow the camera series $\mathbf{C} = \{c^i\}_{i=1}^{n_{\text{sample}}}$ in real-world self-portrait datasets (Wang et al., 2023a). This approach ensures that the prior information supports the reconstruction of a high-fidelity 3D head avatar while avoiding excessive style inconsistency. Thus, given sampled camera series $\mathbf{C}$, the 3D-aware priors $\mathbf{X} = \{\hat{x}^i\}_{i=1}^{n_{\text{sample}}}$ can be obtained by Eq. 3.

### 4.3 STYLE CONSISTENCY: CONSISTENCY SCORE DISTILLATION

To create a stylized 3D head avatar, a straightforward method is to stylize each image in 3D-aware priors $\mathbf{X}$, using an image editing method (*e.g.*, InstructPix2Pix (Brooks et al., 2023)). Then, the stylized images can be used to train a NeRF. However, our experiments (Fig. 4, second line) show that directly applying an image editing model on individual images leads to inconsistent edits across viewpoints. As a solution to this, we convert the per-frame image editing into a 3D-consistency generation with image-text guidance by equipping the image editing method (Brooks et al., 2023) with a score distilling-based loss (Poole et al., 2022; Wang et al., 2023b). It progressively updates the 3D scene and aligns the editing through a 3D-aware representation, such as NeRF (Mildenhall et al., 2020; Müller et al., 2022). In this way, the editing across views will be aligned in a 3D space, leading to a style consistency generation.

Despite yielding promising results in the generation of 3D objects, VSD loss (Wang et al., 2023b) operates without considering image condition, leading to an absence of content consistency. Consequently, we propose a consistency score distillation (CSD) loss, which could handle the image condition and maintain style consistency. To elaborate, given a style prompt $y$, portrait image $\tilde{x}$ in camera pose $c$, we assume there exists a probabilistic distribution of all possible 3D representations. Under a 3D representation (*e.g.*, NeRF), such a distribution can be modeled as a probabilistic density $\mu(\theta|y, \tilde{x})$. We optimize the distribution to align with the one defined by the image editing model with text-image guidance. Therefore, CSD loss is defined as the score difference between the conditional editing model and the rendered image, and its gradient is given by:

$$\nabla_\theta \mathcal{L}_{\text{CSD}}(\theta) = \mathbb{E}_{t,\epsilon,c}[w(t)(\epsilon_{\text{pretrain}}(x_t; g_{\text{prior}}(c; z_{\tilde{x}}), y, t) - \epsilon_\phi(x_t; g_{\text{prior}}(c; z_{\tilde{x}}), y, t, c))\frac{\partial x}{\partial \theta}], \tag{4}$$

where $g_{\text{prior}}(c; z_{\tilde{x}})$ is the reference image, conditioned on the $z_{\tilde{x}}$ solely to guarantee the content consistency (Sec. 4.2). Compared to the VSD loss, the CSD loss incorporates camera conditions in both score estimators to enhance the 3D awareness. $\epsilon_\phi(x_t; g_{\text{prior}}(c; z_{\tilde{x}}), y, t, c)$ is the LoRA of the pretrained model used to estimate the distribution of rendered image. In training process, the target of LoRA can be formulated below:

$$\min_\phi \mathbb{E}_{t,\epsilon,c}[w(t)||\epsilon_\phi(x_t; g_{\text{prior}}(c; z_{\tilde{x}}), y, t, c) - \epsilon||_2^2]. \tag{5}$$

At inference, $\epsilon_{\text{pretrain}}$ relies on Classifier-free Guidance (CFG) (Ho & Salimans, 2022), which allows higher quality sample generation by introducing additional parameter $w_s$ and $w_y$ as follows:

$$\begin{aligned}
\epsilon^{w_s, w_y}(x_t; \tilde{x}, y, t) = {}& \epsilon(x_t; \varnothing, \varnothing, t) + w_s(\epsilon(x_t; \tilde{x}, \varnothing, t) - \epsilon(x_t; \varnothing, \varnothing, t)) \\
& + w_y(\epsilon(x_t; \tilde{x}, y, t) - \epsilon(x_t; \tilde{x}, \varnothing, t)),
\end{aligned} \tag{6}$$

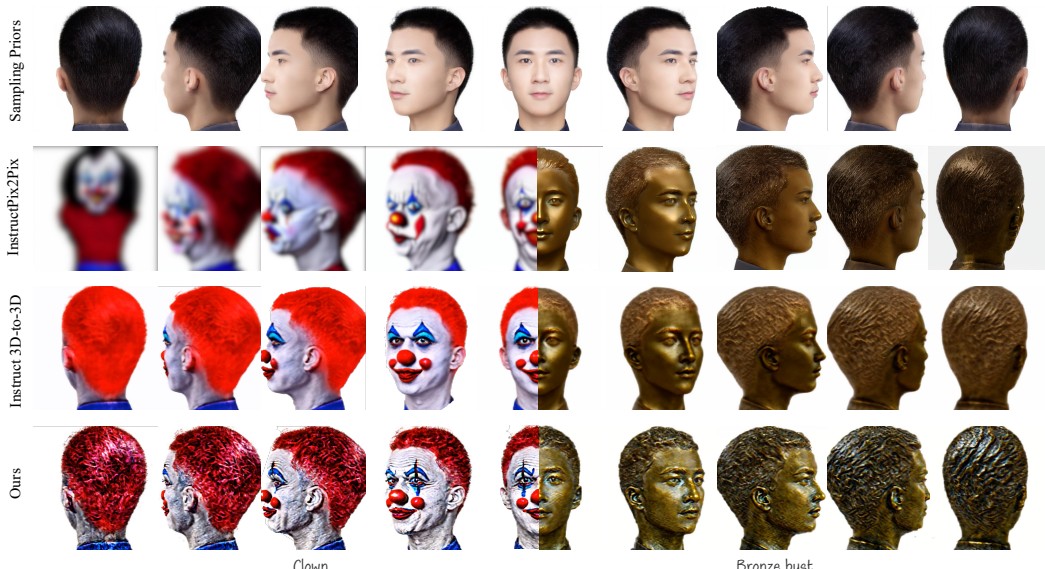

Figure 4: **Baseline Comparisons:** We compare our method with a collection of variants. The priors synthesized through 3D GAN are shown in the first line. Compared with the InstructPix2Pix (per-frame edit) (Brooks et al., 2023) and InstructPix2Pix-based 3D editing methods (Instruct-NeRF2NeRF (Haque et al., 2023), Instruct 3D-to-3D (Kamata et al., 2023)), our methods show a content-style consistency generation with richer details. Some results are blurred for a better view.

where $w_s$ and $w_y$ are the CFG parameter that controls the guidance strength of style prompt $y$ and portrait image $\tilde{x}$, respectively. Additionally, $\epsilon_\phi$ could be formulated as below:

$$\epsilon_\phi(x_t; \tilde{x}, y, t, c) = \epsilon_\phi(x_t; \tilde{x}, y, t, \varnothing) + w_c(\epsilon_\phi(x_t; \tilde{x}, y, t, c) - \epsilon_\phi(x_t; \tilde{x}, y, t, \varnothing)), \quad (7)$$

where $w_c$ is an additional parameter which controls the strength of camera condition.

In practice, we parameterize $\epsilon_{\text{pretrain}}$ by a diffusion-based image editing model, Instruct-Pix2Pix (Brooks et al., 2023). InstructPix2Pix is an instruction-based editing method, which is fine-tuned based on Stable Diffusion (Rombach et al., 2022) and Prompt-to-Prompt (Hertz et al., 2022). Compared with conditional-guided diffusion models, such as ControlNet (Zhang et al., 2023d), InstructPix2Pix demonstrates the ability to edit images while maintaining camera consistency and preserving identity information more effectively.

### 4.4 STYLEDREAMER

As discussed above, we encourage a 3D avatar generation of consistency in both content and style, via 3D-aware priors sampling and consistency score distillation, respectively. Then, we introduce the generation pipeline in detail.

**1. 3D-aware priors sampling.** Given a portrait image $\tilde{x}$ and a camera set $\{c^i\}_{i=1}^{n_{\text{sample}}}$, we first collect the 3D-aware priors $\mathbf{X}$ and corresponding camera set $\mathbf{C}$. We crop and obtain the camera poses $\tilde{c}$ of portrait image using 3DDFA (Guo et al., 2020). Then, we compute the latent code $z$ with PanoHead An et al. (2023) using $\tilde{x}$. Moreover, the 3D-aware priors $\{\hat{x}^i\}_{i=1}^{n_{\text{sample}}}$ can be obtained by Eq.(3) with sampled camera series $\{c^i\}_{i=1}^{n_{\text{sample}}}$. The 3D-aware priors $\tilde{\mathbf{X}}$ is the union of $\tilde{x}$ and sampled priors $\{\hat{x}\}$, and the camera set $\mathbf{C}$ can be obtained in the same way.

**2. Rendering.** Randomly selecting a camera pose in $C$, we render the NeRF at $512 \times 512$ resolution using the hash grid encoding from Instant NGP (Müller et al., 2022). It allows us to represent high-frequency details at a much lower computational cost.

**3. CSD loss.** We use the annealed time schedule (Wang et al., 2023b) for score distillation. For the first 5,000 iterations, we sample time step $t \sim \mathcal{U}(0.02, 0.98)$ and then anneal into $t \sim \mathcal{U}(0.02, 0.50)$. For classifier-free guidance, we set $w_s = 7.5$, $w_y = 3.5$ and $w_c = 1.0$, finding that higher image guidance weights can reduce the Janus problem. Given the rendered image, sampled timestep t,

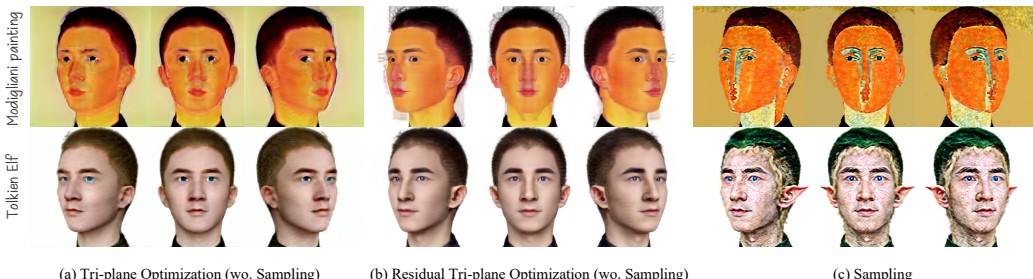

(a) Tri-plane Optimization (wo. Sampling)  (b) Residual Tri-plane Optimization (wo. Sampling)  (c) Sampling

Figure 5: **Sampling:** Directly editing the 3D representation (Tri-plane with NeRF) in 3D GAN is hard. Therefore, we choose to sample multiple views from it to train a new NeRF.

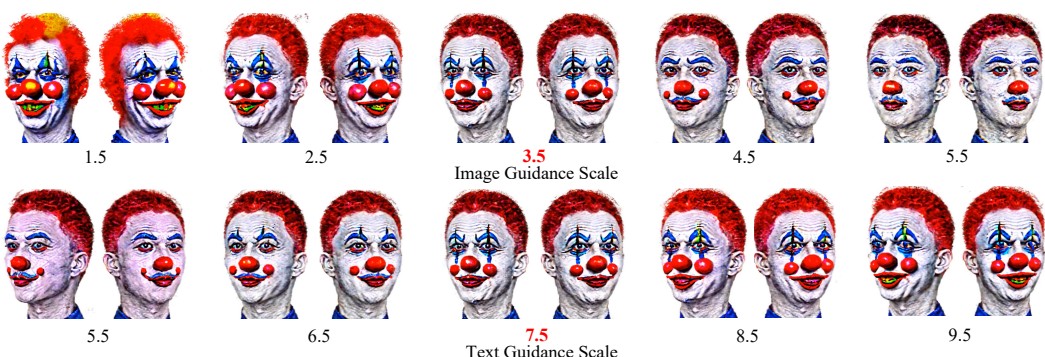

Figure 6: **Guidance Scale:** We validate the effectiveness of CSD with different image guidance scale $w_s$ and text guidance scale $w_y$ values, respectively. The guidance scale in red indicates the default setting in our paper.

image guidance $\tilde{x}$, style prompt $y$ and camera $c$, we compute the gradient of the NeRF parameters according to Eq.(4). Meanwhile, the LoRA is also updated on the rendered image according to Eq.(5). Additionally, the normal vector regularization (Melas-Kyriazi et al., 2023; Lin et al., 2023a) is introduced to enforce the smoothness of the normal maps of geometry. Meanwhile, a $\mathcal{L}_1$ and LPIPS loss (Zhang et al., 2018b) are used in the first 2,000 iterations to help the NeRF's convergence.

**4. Optimization.** A stylized 3D head avatar is trained on one GPU (24GB GeForce RTX 4090), where the batch size is 1. We optimize for 20,000 iterations which takes around 1.5 hours. Parameters are optimized using the Adam optimizer (Kingma & Ba, 2014). More optimizations settings can be found in Appendix.

## 5 EXPERIMENTS

We conduct experiments on One-to-Style dataset using threestudio (Guo et al., 2023). First, we investigate the benefits of proposed pipeline with qualitative experiments. Second, various ablation study are designed to discuss our choices in proposed pipeline.

### 5.1 QUALITATIVE EVALUATION

**Head avatar generation with various style prompts.** As shown in Fig. 2, 3D head avatars in diverse styles are generated by our methods, demonstrating high-quality geometry and texture in full views. Our method is emphasized by its ability to create head avatars of ordinary individuals and handle multiple styles, including artistic style (*e.g.*, Ddward Munch painting), material style (*e.g.*, marble statue) and character style (*e.g.*, clown). It is worth noting that our method shows an ability for fine-grained semantic understanding. In specific, when stylizing the avatar with Van Gogh's Starry Night, only the hair is stylized in Starry Night style rather than in a global way.

**Comparison with variants.** We validate our design choices by comparing our approach to the following variants. The qualitative differences are shown in Fig. 4.

*Per-frame Edit.* The most naïve baseline related to our is to apply InstructPix2Pix (Brooks et al., 2023) on every sampled priors (Fig 4, second line). Distortion are observed in these images when faces are in large pose. Specifically, in the " clown " case, the back of the head suffers from the Janus problem (Hong et al., 2023). Meanwhile, both content and style are inconsistent across views.

*SDS + InstructPix2Pix.* This method (Kamata et al., 2023) is an InstructPix2Pix-based 3D editing method. We realize it by equipping the InstructPix2Pix with SDS loss (Poole et al., 2022) and let the sampled prior image a condition. Specifically, we calculate its gradient by the difference between InstructPix2Pix's predicted noise and added noise. As shown in the fourth line of Fig. 4, the generated head avatars suffer from over-saturation and over-smoothing problems. These problems are especially reflected in the hair of the "clown" case, where red dominates the back of the head area resulting in poor texture.

### 5.2 ABLATION STUDY

In this part, we try to answer the following questions: **Q1.** how CFG parameters affect the head avatar generation? **Q2.** why do we train a new NeRF rather than directly edit the 3D representation extracted in 3D GAN?

**A1.** The image guidance scale and text guidance scale control the content and style of the generated head avatar, respectively. Figure 6 shows the differences by changing the image guidance scale $w_s$ and text guidance scale $w_y$. The results show that using a small image guidance scale (e.g., 1.5) with a relatively large text guidance scale (e.g., 7.5) will lead to an over-stylization problem. This problem reflects the loss of identity (ID) and further introduces the Janus problem (Hong et al., 2023), characterized by multiple faces in a head. Therefore, we choose a relatively large $w_s$ to preserve the content of the person.

**A2.** Directly edit the 3D representation in 3D GAN is hard. As described in Sec. 4.2, we apply GAN inversion (Roich et al., 2022) on PanoHead (An et al., 2023) to reconstruct a 3D face of the given portrait image. Based on this, the 3D face can be represented by a combination of a tri-grid and a shared NeRF. Ideally, editing based on this tri-grid can not only preserve the information but also speed up convergence. Therefore, to explore this problem, we designed two toy experiments: tri-grid optimization and residual tri-grid optimization.

For tri-grid optimization, we fix the parameter of shared NeRF and edit the tri-gird with CSD loss directly. However, as shown in Fig. 5 (a), only the color of the head avatar is edited. Meanwhile, the identity of the head avatar is broken. Furthermore, to preserve the identity of the head avatar, we fix the tri-grid and initialize a new tri-grid to learn the editing residual. The results are shown in Fig. 5 (b), the geometry is still hard to change. We infer it is because the parameter space of the tri-grid is huge, resulting in hard convergence. Additionally, with the development of NeRF, a 3D scene could be created quickly. Therefore, in this paper, we propose to sample novel views as reference images and train a new NeRF.

## 6 CONCLUSION

In this paper, we introduce a new task, One-to-Style, to generate personalized head avatars for users. It poses challenges in maintaining content and style consistency. To overcome this, we propose StyleDreamer, a new generation pipeline. It encourages consistency in content and style through 3D-aware priors sampling and consistency score distillation. Specifically, we synthesize novel views using a pre-trained 3D GAN for content consistency. Then, we use CSD loss, a score distilling-based loss, to align cross-view editing in a 3D space for style consistency. We present results on various portrait images and styles, highlighting its potential real-world application.

**Limitations.** Although our method produces impressive results, it inherits limitations from PanoHead (An et al., 2023) and InstructPix2Pix (Brooks et al., 2023). For instance, InstructPix2Pix is unable to perform large spatial manipulations, and PanoHead struggles with complex backgrounds and large-pose faces, leading to artifacts in face reconstruction. This problem impacts the quality of sampled priors and degrades the performance of 3D stylization.

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
