# Appendix

## A  INTRODUCTION

Our appendix contains further details about our research. The content is organized as follows:

- In Section B, we provide more information aboud benchmark dataset of One-to-Style task.
- In Section C, we provide more trianing details of StyleDreamer.

## B  BENCHMARK DATASET OF ONE-TO-STYLE

We first present out the portrait images and style prompt in benchmark dataset. Fig. 7 is the portrait images in evaluation dataset. Tab. 1 shows the style prompt we used in evaluation.

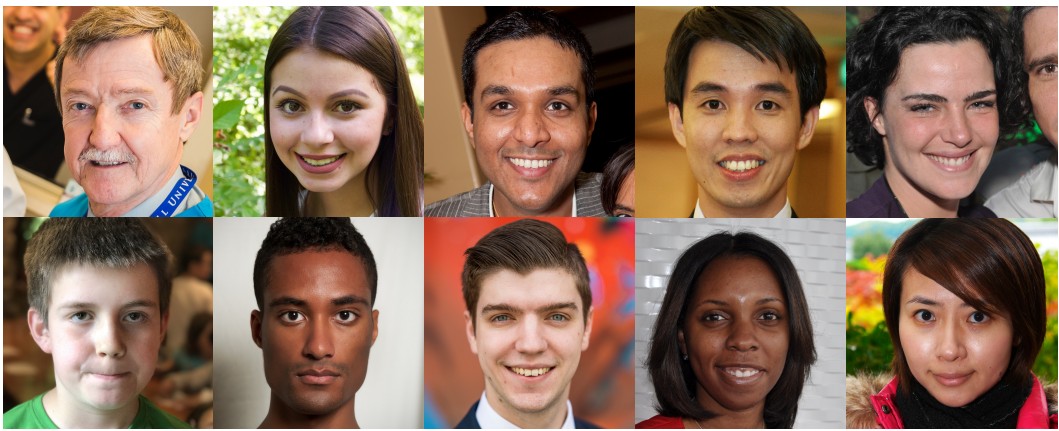

Figure 7: **Portrait Images in Benchmark Dataset:** The portrait images in evaluation dataset of One-to-Style task.

| Index | Style | Prompt |
|---|---|---|
| | *Artistic Style* | |
| 1 | Van Gogh's Starry Night | Transform it to Van Gogh, Starry Night style |
| 2 | Modigliani painting | Transform it to Modigliani painting style |
| 3 | Fauvism Painting | Make him look like a Fauvism painting |
| 4 | Edvard Munch Painting | Make him look like an Edvard Munch painting |
| 5 | Andy Warhol Painting | Transform it to Andy Warhol painting |
| 6 | Pixar | Turn him into the style of Pixar |
| 7 | Disney | Turn him into the style of Disney |
| | *Material Style* | |
| 8 | Bronze Bust | As a bronze bust |
| 9 | Marble Statue | Make him a marble statue |
| | *Character Style* | |
| 10 | Tolkien Elf | Turn him into the Tolkien Elf |
| 11 | Clown | Turn him into a clown |
| 12 | Voldemort | Turn him into a Voldemort |
| 13 | Anime | What if he were an anime character |
| 14 | Robot | Turn him into a robot |
| 15 | Cyborg | Turn him into a cyborg |

Table 1: **Style Prompts in Benchmark Dataset:** The style prompts in evaluation dataset of One-to-Style task.

## C  TRAINING DETAILS OF STYLEDREAMER

As shown in Tab. 2, we present the details of loss wight used in training and the learning rate of trained module.

| Optimizer Parameters | |
|---|---|
| *Loss* | *Weight* |
| CSD (Eq. 4) | 1 |
| LoRA (Eq. 5) | 1 |
| $\mathcal{L}_1$ | 10 |
| LPIPS | 10 |
| Normal Vector Regularization | 1000 |
| *Module* | *Learning Rate* |
| NeRF | 0.01 |
| LoRA-geometry | 0.01 |
| LoRA-background | 0.001 |
| LoRA-guidance | 0.001 |

Table 2: **Optimizer Parameters:**  The details of loss wight used in training and the learning rate of trained module.