# OpenReview forum: "StyleDreamer: Make Your 3D Style Avatar from a Single View with Consistency Score Distillation"
_ICLR.cc/2024/Conference — ICLR 2024 Conference Withdrawn Submission_

### Official Review · Reviewer_cZ1J · 2023-10-29

**Soundness:** 3 good
**Presentation:** 3 good
**Contribution:** 1 poor
**Rating:** 3
**Confidence:** 4

**Summary:**

This paper proposes a new framework for generating stylized heads based on a single image. To convert the image into an avatar, two submodules are introduced for image-to-nerf and nerf stylizing. For the first module, a gan-based 360-degree head image synthesis method is adopted to convert the single-view image to multi-view head images for initializing the corresponding nerf. In the next module, the instruct pix2pix method is used with a LORA subnet for generating high-quality stylized nerf. Experimental results demonstrate that the input images are successfully converted into stylized 3D nerf-based avatars.

**Strengths:**

+ This paper is well-organized and clearly explains the process of converting a single image into multi-view images and the styling of a Nerf.

+ The proposed framework is technically sound, where a Lora is utilized to enhance generation quality.

+ The experimental results demonstrate that the new framework can generate stylized Nerfs with richer details.

**Weaknesses:**

- The contribution of this work is limited. Although a new benchmark dataset is presented as a contribution, it contains very few images and styles, which makes this benchmark unconvincing.

- The proposed method is quite incremental. This paper simply combines four previous works, including PanoHead, Instant-NGP, Instruct Pix2Pix, and ProlificDreamer, while the main improvement (i.e., the richer details shown in Fig. 4) is attributed to the VSD loss.

- The experiment in Fig. 4 is not fair enough. The Nerf used in Instruct Pix2Pix is DVGO, while the Nerf used in this work is Instant NGP, which may introduce extra bias into the comparison.

**Questions:**

1. Why not include more images and styles into the new benchmark dataset? For example, only one photo of elderly people is included in this dataset, which is far from enough to represent the diversity of this group.

2. Please provide more comparison results, since the Fig.4 only provides one identity and two styles. Video results are also encouraged.

3. What would happen if the DVGO is adopted in this work?

---

### Official Review · Reviewer_Sr2S · 2023-10-31

**Soundness:** 3 good
**Presentation:** 3 good
**Contribution:** 3 good
**Rating:** 5
**Confidence:** 2

**Summary:**

The paper proposes a method for 3D Avatar generation using a text prompt captured from a single angle. The argument that the authors put forward is that multi-view data is hard to acquire which warrants the need for methods that work on a single image. In this context, the paper focuses on two types of consistencies i) content consistency, which is the method proposed to solve by using a 3D GAN inversion method that projects a single aligned image into 3D using the 3D-GAN latent spaces. This renders a view-consistent 3D model of the face. ii) Style consistency, which uses diffusion-based consistency score distillation loss to maintain the consistency of the styles in different views. The paper shows the result of 3D avatars under different artistic prompts throughout the paper.

**Strengths:**

1) The paper focuses on a single image to stylized 3D avatar generation, which is a challenging task compared to a multi-image 3D stylization task. The multi-image 3D Stylization assumes the availability of such data, which is difficult to acquire in the real-world scenario. Moreover, techniques developed on top of such data may very well overfit and not generalize.

2) The paper shows the results using different artistic prompts throughout the paper. The results show that the output respects the content in the prompts and renders the results at a higher resolution without the blurry artifacts as noticeable in the competing works.

3) The method can handle geometric deformations like elf ears. This shows that the method can produce additional geometric details than just painting textures on the 3D model it is optimizing.

**Weaknesses:**

1) The paper falls short in providing quantitative comparisons with the methods it references. While the paper discusses relevant metrics, it lacks a clear presentation of results in tabular form. A comprehensive quantitative analysis is essential to assess the method's quality. Additionally, considering the subjective nature of stylization, a user study could be valuable.

2) The paper assumes the availability of a 3D GAN to generate multiple views, which is suitable for the specific scenario described. However, this assumption may not hold for other datasets, particularly in cases such as full-body humans, where maintaining quality and multiview consistency can be challenging.

3) Multiview consistency appears compromised, as indicated by the supplementary videos. The results in Figure 1 and the accompanying videos show graininess. Furthermore, the normal maps also show this graininess.

4) The framework lacks a dedicated module explicitly designed to preserve identity, as suggested in the introduction. Although a 3D GAN may be capable of producing view-consistent images, there is no guarantee that the stylized output can preserve the identity of the subjects.

5) Despite the use of an editable 3D GAN, the avatars generated by the method are not editable. Instead, the method trains a separate Nerf, which limits the manipulation abilities inherent to a 3D GAN, both in terms of local and global edits to images.

**Questions:**

1) Regarding the lack of quantitative comparisons, could you provide insights into why there are no tables showing results in the paper despite discussing metrics? How do you plan to address this limitation, and do you have plans for a quantitative analysis or a user study to assess the method's performance more rigorously?

2) The paper assumes the availability of a 3D GAN for generating multiple views. How do you anticipate this assumption impacting the applicability of your method to datasets where this assumption may not hold, particularly in cases involving full-body humans?

3) For the concerns about multiview consistency and the graininess observed in results and normal maps, could you explain potential reasons for these issues and any steps taken to mitigate them?

4) The paper mentions identity preservation in the introduction, but there doesn't appear to be an explicit module designed for this purpose. Could you elaborate on how the method aims to maintain identity throughout the stylization process, especially after using a 3D GAN as a base?

5) Can the outputs be editable?

---

### Official Review · Reviewer_hk3d · 2023-10-31

**Soundness:** 3 good
**Presentation:** 3 good
**Contribution:** 2 fair
**Rating:** 5
**Confidence:** 4

**Summary:**

The paper proposes a way to generate a stylizable 3D head from a single image. The method utilizes 3D-GAN inversion to obtain a 3D head representation from the input image. A text-guided nerf-based representation is then trained using the prior and a text prompt. To this end, the authors utilize an instant-NGP-based representation and a score distillation-based loss. The qualitative results highlight the efficacy of the method.

**Strengths:**

- The application of generating stylazable heads from a single image is interesting and relevant.
- The qualitative results indeed show that the approach outperform the baselines in terms of visual quality.
- I appreciate the ablation study on the need for a separate NeRF.

**Weaknesses:**

- The main issue of the paper is the lack of novelty. Although I do believe that a method that combines SOTA components to solve a novel problem is sufficient, the proposed task is not a sufficient contribution in itself.
- In the same vain, I believe the contribution of the benchmark is a bit overstated. The size of the benchmark is not sufficient and the paper lacks proper evaluation of different methods. This would be vital to evaluate the contribution of a benchmark.
- I am missing a comparison to text-guided head models, even explicit ones, e.g., ClipFace [1]
- The authors proposed a quantitative evaluation protocol, however I am missing a table comparing the baselines.



[1] Text-guided Editing of Textured 3D Morphable Models

**Questions:**

I suggest that the authors soften the contribution claims and add further comparison to more baselines as well as quantitative evaluation.

---

### Official Review · Reviewer_7NEF · 2023-11-06

**Soundness:** 2 fair
**Presentation:** 3 good
**Contribution:** 2 fair
**Rating:** 3
**Confidence:** 5

**Summary:**

This paper proposes a method called StyleDreamer, to lift a provided 2D image of a face into 3D and then to create a stylized version of the 3D representation of the face conditioned on a text prompt. The proposed method involves first using PanoHead with Pivotal tuning to create a 3D tri-grid representation of the provided face. Then, multiple views of the face are rendered and input into InstructPix2Pix to learn an InstantNGP-style Nerf representation for the face using a proposed CSD loss. The CSD loss guides image generation based on the scores of the reference image, the input text prompt and the camera viewpoint using the variation score distillation loss (VSD) + LoRA formulation of ProlificDreamer. The authors provide various qualitative results for comparisons of their method to some alternative approaches and ablations of their method.

**Strengths:**

Originality: The paper proposes a new method for what the authors call the task of "One-to-Style", that is of simultaneously lifting a 2D face into 3D and of stylizing it with a text prompt. They also propose a new method, which combine various several state-of-the-art models to solve this task.

Clarity: The presentation of the technical details of the method and the overall organization of the paper are good.

**Weaknesses:**

1. Originality: The authors claim "Despite promising results, these methods are unable to generalize to out-of-domain styles since they solely
concentrate on one style during training." This is not entirely true. In Trevithick et. al. Real-Time Radiance Fields for Single-Image Portrait View Synthesis, SIGGPRAH 2023, Figure 8., which proposes an encoder into EG3D, many examples of out-of-domain generalization for stylized facial images of are shown. A straight-forward alternative baseline to the proposed approach in this paper is then to first use InstructPix2Pix on 2D facial images to stylize them with a prompt and then to lift them into a 3D facial GAN's latent space (EG3D, PanoHead) via an encoder or GAN inversion. The authors don't show any results of actually doing this, but simply claim that it doesn't work. In my experience, this also works fine and produces much better texture and shape quality. So I am not entirely convinced that the proposed approach is needed.

2. Results: We results of the proposed method are not convincing for the following reasons:
(a) In general the quality of the learned texture and normals is very noisy.
(b) Only qualitative results of a few cherry-picked examples are shown. No aggregate quantitative results via a user-study or using the proposed ID Score, and Style Score are shown.
(c) The curated evaluation dataset of 10 subjects, presented in the supplementary document are all from the FFHQ dataset, which was used to train PanoHead. Ideally the evaluation dataset should not comprise of training images, to understand the ability of the proposed method to generalize to faces not seen during training.
(d) An ablation of the contribution of the "camera" conditioning in the proposed StyleDreamer is not presented.
(e) An ablation of using InstructPix2Pix with VSD instead of SDS is not presented.
(f) The caption for Figure 4, says that results of Instruct Nerf2Nerf are also present, but the visual results are absent.

3. Significance: Overall the quality of the results is not sufficiently high enough to warrant acceptance. The value of stylizing a face image in the style of "Starry night" is of limited value.

**Questions:**

The authors mention that optimizing the tri-grid doesn't work and hypothesize that this happens because the tri-grid has too many parameters. Did they try with EG3D and its tri-plane? In our experiments, we've experimentally verified that tri-grid is not required for full 360 rendering and both methods suffer from artifacts when rendering the back view. The back view rendering issue stems more from the lack of training data from the back view versus any fundamental limitation of the tri-plane representation.

I would also like to hear the authors' responses to the three weaknesses that I've pointed out in the previous section.